# Association between Level of Empathy, Attitude towards Physical Education and Victimization in Adolescents: A Multi-Group Structural Equation Analysis

**DOI:** 10.3390/ijerph16132360

**Published:** 2019-07-03

**Authors:** Víctor Arufe-Giráldez, Félix Zurita-Ortega, Rosario Padial-Ruz, Manuel Castro-Sánchez

**Affiliations:** 1Department of Specific Didactics and Methods of Research and Diagnosis in Education, University of Coruña, 15001 Coruña, Spain; 2Department of Didactics of Musical, Plastic and Corporal Expression, University of Granada, 18071 Granada, Spain

**Keywords:** victimization, empathy, physical education, adolescent, physical activity

## Abstract

Currently, there is a problem related to violence and tolerance towards violence, and the levels of empathy of the subjects can affect this, in addition to the practice of physical activity. The present study sought to define and contrast an explanatory model of victimization, empathy and attitude towards physical education, and to analyze the existing relationships between these variables as a function of engagement with physical activity. A total of 2388 adolescents from Spain participated in this research. The sample was made up of both sexes (43.39% males and 56.61% females), with ages reported between 11 and 17 years (M = 13.85; SD = 1.26). Empathy (TECA), attitude towards physical education (CAEF) and victimization (EV) were measured. A multi-group structural equation model was developed, which showed excellent fit to the empirical data (χ^2^ = 559.577; DF = 38; *p* < 0.001; comparative fit index (CFI) = 0.957; normalized fit index (NFI) = 0.954; incremental fit index (IFI) = 0.957; root mean squared error of the mean (RMSEA) = 0.054). A direct positive relationship exists between affective and cognitive empathy. A positive association was found between motivational climate and engagement in physical activity. With regards to victimization, the verbal dimension obtained the highest correlation score, followed by the relational and physical dimensions. In the case of adolescents who regularly engaged in physical activity, the verbal and physical dimensions presented the weakest correlations, whilst the relational dimension was the most strongly associated in the case of sedentary adolescents. The main conclusions of the present study it is that the relationship between affective empathy and cognitive empathy is positive and direct, however, this relationship is slightly stronger and differentiated in sedentary adolescents than in those who practice physical activity on a regular basis.

## 1. Introduction

At present, violent behaviors are more visible, due to the information society, with a multitude of news media related to violent behavior [1,2,3]. In the academic environment, in the stages of primary and secondary education, various violent situations occur among students. For these reasons, today’s society is more sensitive to situations of violence [4]. Adolescence represents a critical period when it comes to acquiring proactive behaviors, while disruptive behaviors can model the adolescent’s personality, because at this stage of life they are very malleable [5]. Therefore, the problem worsens for the victim in cases of violence, because these subjects suffer the direct consequences of this problem, in some cases leading to depression and even suicide [6].

Research conducted on violent behavior in the academic field has increased during the last years, finding that families and educational agents can greatly influence the development and tolerance of this type of violent behavior [7,8,9,10,11,12,13]. Amongst the elements for reducing these harmful behaviors, individuals must first identify the problem so that they can then create behavioral patterns which will become established in later life stages such as adulthood [14,15].

One of the psychological factors that is most linked to the development of violent behavior and victimization is empathy, understood as the ability to put oneself in the situation of others [16]. Therefore, empathy can be a key factor when it comes to tolerating violent behavior, because subjects with higher levels of empathy will be less likely to tolerate such situations [17,18], while people with less empathy do not. They are able to put themselves in the place of the victim and therefore are more tolerant with situations of peer violence that they perceive [19,20].

Engagement with physical activity contributes significantly to the development of the dimensions of the human personality. These include emotional intelligence, self-conceptualization and the social factors that help individuals to relate themselves with others [21]. School tends to be the medium most used by teachers and other educational establishments when trying to develop prevention programs targeting violent behaviors.

In this respect, it is considered essential during the adolescent stage to engage in physical activity for one hour a day [22], as regularly engaging in physical activity appears to be central to the prevention and treatment of a number of psychological and physiological problems [23,24]. Further, it has been determined that regular exercise produces a type of socialization that is highly interesting as a subject of research. In this sense, the development of physical activity offers an excellent medium for the reduction of violent conduct [13,25].

Based on the scarcity of studies available that examine empathy, physical activity and victimization, the following study was proposed with the objective of defining and contrasting an explanatory model of empathy, attitude towards physical education, and victimization in adolescents. A further objective was to analyze, through multi-group structural equation analysis, the existing associations between empathy, attitude towards physical education and victimization as a function of whether or not physical activity was regularly engaged in.

## 2. Materials and Methods

### 2.1. Design and Participants

A total of 2388 adolescents enrolled in compulsory secondary education participated in the present descriptive and cross-sectional study. Ages were reported between 11 and 17 years (M = 13.85 years old; SD = 1.268), with 1036 (43.39%) being male and 1352 (56.61%) being female. Sample selection was conducted through a process of convenience sampling attending to criteria for compulsory secondary education. Inclusion criteria required participants to provide informed consent from their parents or legal guardians and to not suffer from any type of pathology that would impede participation in the research. The sample was obtained from eight Spanish cities, requesting participation from all centers who voluntarily accepted to collaborate. It is necessary to indicate that 281 questionnaires were excluded after detecting that they had been incorrectly completed or that data was missing from them. The initial sample consisted of 2669 adolescents, and finally, after eliminating the questionnaires invalidated by their incorrect completion, the total sample was 2388 adolescents who answered the questionnaire correctly. The response rate was 89.47%. For the selection of participants, the only criteria used was that the adolescents attended compulsory secondary education and did not have any impediment that prevented them from participating in the study. An informed consent form addressed to parents or legal guardians was used because the participants were minors. 

### 2.2. Variables and Instruments

Regular physical activity practice. Collection by means of an ad hoc questionnaire in which the subject is asked if the subject performs three or more hours of physical activity a week. Dichotomic answer: “Yes” or “No”. It is considered that a subject who practices physical activity three or more hours a week out of school is physically active, therefore this criterion has been used, coinciding with international recommendations.Cognitive and Affective Empathy Test (TECA). The original questionnaire developed by López-Pérez, Fernández-Pinto and Abad [26] was used. This seeks to analyze participants’ capacity for empathy from a dual perspective, cognitive and affective. The questionnaire was composed of 33 items that were scored according to a five-point Likert scale. The instrument presents a three factor model, composed of the two large dimensions of cognitive empathy and affective empathy. These in turn were each composed of two factors each. Cognitive empathy was formed by perspective adoption (capacity of the individual to communicate themselves, tolerate the rest and maintain social relationships) and emotional understanding (ability to recognize and understand the emotions and impressions of others). Affective empathy was formed by empathetic stress (ability to understand and share the negative emotions of others) and empathetic joy (capacity to understand and share the positive emotions of others). A reliability index of α = 0.80 was obtained in the present study, this being similar to that found in the study conducted by López-Pérez et al. [26], in which a reliability index of α = 0.82 was established.Attitudes towards Physical Education Questionnaire (CAEF). The original questionnaire described by Moreno, Rodríguez and Gutiérrez [27] was used. The questionnaire is composed of 56 items rated along a four-point Likert scale, where 1 = disagree and 4 = totally agree. This instrument is composed of seven different dimensions: evaluation of the subject matter and of the Physical Education teacher, difficulty of Physical Education, usefulness of Physical Education, empathy with the teacher and the subject matter, agreement with organization of the subject matter, preference for Physical Education and sport, and Physical Education as sport. For this instrument a consistency of α = 0.75 was obtained, with this being acceptable and slightly higher than the value obtained by Moreno et al. [27] in their original study (α = 0.73).Victimization at School Scale (EV). The present instrument was created by Mynard and Joseph [28] and adapted into Spanish by Cava, Musitu and Murgui [29]. The questionnaire is composed of 20 items evaluated according to a four-point Likert scale, where 1 = never and 4 = always. This instrument provides information about three types of victimization: physical, verbal and relational. In the original study, Mynard and Joseph [28] obtained an internal consistency (Cronbach alpha) value of α = 0.77. In the present study, a superior coefficient of α = 0.93 was obtained, with this being similar to that obtained by Povedano, Estévez, Martínez and Monreal [30].

### 2.3. Procedure

Educational centers were contacted by the University of Granada to inform them about the nature of the study, proceeding to select the centers who voluntarily agreed to participate. Documents of informed consent were administered to the pupils of the center, requesting permission for participation from parents or legal guardians. Following this, the questionnaires were administered in groups during school hours. Anonymity of participants was guaranteed, stipulating that all data collected would only be used for scientific purposes. Researchers were present during data collection in order to guarantee the correct development of the process and to resolve any doubt. The research received approval from the ethics committee of the University of Granada, project code 641/CEIH/2018.

### 2.4. Data Analysis

For basic descriptive analysis, the statistical software IBM SPSS® was used in version 22.0 (IBM Corp., Armonk, NY, USA) for Windows. The program IBM AMOS® 23 (IBM Corp., Armonk, NY, USA) was employed with the purpose of analyzing the existing associations between the constructs implied in the structural model. After developing the theoretical model, path analysis was conducted considering the correlation matrix through multi-group analysis, grouping participants according to those who engage regularly in physical activity and those who don’t. Finally, two different structural models were configured with the aim of verifying whether the relationships between the studied variables varied as a function of adolescents’ level of engagement in physical activity.

The models produced from the path analysis are made up of nine observable variables and three latent variables in order to determine the indicators (Figure 1). The models proposed formulate causal explications for the latent variables through the associations observed between the indicators, whilst taking into consideration the reliability of the measures. Further, the observable variables include the relevant measurement errors with the aim of directly controlling them. Unidirectional arrows are lines of influence between the latent and observable variables, being interpreted as multivariate regression coefficients. Bidirectional arrows show the relationship between latent variables, also representing regression coefficients.

Cognitive empathy (EC) and affective empathy (EA) act as exogenous variables, with each one being inferred by two indicators. With regards to cognitive empathy, the indicators are AP (perspective adoption) and CE (emotional understanding). For affective empathy, the indicators are EE (empathetic stress) and AE (empathetic joy). Evaluation of the subject and of the Physical Education teacher (VEF) acts as an endogenous variable, receiving the effect of cognitive empathy (EC) and affective empathy (EA). Preference for Physical Education and sport (PEFD) acts as an endogenous variable, receiving the effect of cognitive empathy (EC), affective empathy (EA) and evaluation of the subject and of the Physical Education teacher (VEF). Further, victimization (V) acts as an exogenous variable that is inferred by three indicators: VR (relational victimization), VF (physical victimization) and VV (verbal victimization). It also acts as an endogenous variable, receiving the effect of cognitive empathy (EC), affective empathy (EA), evaluation of the subject and of the Physical Education teacher (VEF), and preference for Physical Education and sport (PEFD).

Model fit was checked with the purpose of verifying compatibility of the model to the empirical data obtained. Reliability of model fit was analyzed according to goodness of fit criteria [31].

## 3. Results

The structural equation model proposed according to physical activity engagement carried out on a sample of 2388 adolescents from the province of Granada reveals a good fit for all evaluation indices. Chi-squared analysis shows a significant *p*-value (χ^2^ = 559.577; DF= 38; *p* < 0.001). However, this index cannot be interpreted in a standardized way, a further problem is that it is highly sensitive to sample size [31]. As a result, other indices of standardized fit which are less sensitive to sample size are employed. Comparative fit index (CFI) obtained a value of 0.957, with this being excellent. Normalized fit index (NFI) produced a value of 0.954 and the incremental fit index (IFI) was 0.957, with both of these being excellent. Root mean squared error of the mean (RMSEA) obtained an acceptable value of 0.054.

Figure 2 and Table 1 show the estimated values of the parameters included in the structural model for the adolescents who regularly engage in physical activity. This value must be of an adequate magnitude and have effects that are significantly different from zero. Likewise, improper estimates such as negative variances must not be obtained.

Statistically significant associations are observed at the level of *p* < 0.001 between all categories of empathy and its dimensions, with these being positive and direct. The association between cognitive empathy and affective empathy is significant to the level of *p* < 0.001, this being positive and direct, and showing a high correlation strength (r = 0.821).

When analyzing the influence of the indicators of empathy, statistically significant differences at the level of *p* < 0.001 were found, with all of these associations being positive and direct. In the case of cognitive empathy, the indicators demonstrated a similar influence, with perspective adoption being the indicator showing the greatest correlation coefficient (r = 0.673), followed by emotional understanding (r = 0.634). With regards to affective empathy, the indicators did not exert a similar influence. Empathetic joy was found to be the indicator with the greatest correlation coefficient (r = 0.617), whilst empathetic stress exerted an influence of a low correlation strength (r = 0.227).

In the consideration of victimization, statistically significant associations at the level of *p* < 0.001 are seen for all dimensions, these being positive and direct. When analyzing the influence of the indicators, the greatest influence is found for verbal victimization (r = 0.939), followed by relational victimization (r = 0.861) and finally, physical victimization (r = 0.747) was the indicator that exerted the least influence.

When analyzing the association between empathy and victimization, no association was found with cognitive empathy (*p* = 0.453), nor with affective empathy (*p* = 0.449). Nor was any association found when associating evaluation of the subject matter and of the Physical Education teacher with cognitive empathy (*p* = 0.170), or with affective empathy (*p* = 0.877). The same outcome was produced when analyzing the association of preference for Physical Education and sport with cognitive empathy (*p* = 0.405), and with affective empathy (*p* = 0.478). 

When the association between the preference for Physical Education and sport, and evaluation of the subject and the Physical Education teacher is analyzed, a statistically significant association is found at a level of *p* < 0.001, showing a positive and direct association (r = 0.236).

In the case of the association between preference for Physical Education and sport with victimization, the existence of a statistical association was uncovered at a level of *p* < 0.01, with a positive and direct relationship being observed (r = 0.106).

Finally, an association at the level of *p* < 0.001 is found between evaluation of the subject and of the Physical Education teacher, and victimization, with this being negative and indirect (r = −0.097). 

Figure 3 and Table 2 show the estimated values produced for the parameters included in the structural model for sedentary adolescents. These must be of an appropriate magnitude and the effects should significantly differ from zero. Likewise, improper estimates such as negative variances must not be obtained.

Statistically significant associations at the level of *p* < 0.001 are observed between all categories of empathy and its dimensions, with these being positive and direct. The association between cognitive empathy and affective empathy is significant at the level of *p* < 0.001, with this being positive and direct, and showing a high correlation strength (r = 0.882).

When analyzing the indicators of empathy, significant differences at the level of *p* < 0.001 were found, with all of these associations being positive and direct. In the case of cognitive empathy, the indicators exerted a similar influence, with perspective adoption being the indicator to demonstrate the largest correlation coefficient (r = 0.690), followed by emotional understanding (r = 0.626). When considering affective empathy, the indicators do not seem to have a similar influence. Empathetic joy was found to be the indicator that showed the greatest correlation coefficient (r = 0.505), whilst empathetic stress exercises an influence with a low correlation strength (r = 0.229).

With regards to victimization, statistically significant associations at the level of *p* < 0.001 are observed with all of its dimensions, with all of these being positive and direct. When analyzing the influence of the indicators, a greater influence is found relating to verbal victimization (r = 0.928), followed by relational victimization (r = 0.870). Finally, the indicator that exerts the weakest influence is physical victimization (r = 0.728).

When analyzing the association between empathy and victimization, no association was found with cognitive empathy (*p* = 0.870), nor with affective empathy (*p* = 0.488). Further, no association was uncovered when associating evaluation of the subject and of the Physical Education teacher, with affective empathy (*p* = 0.461). However, when analyzing the association of evaluation of the subject and of the Physical Education teacher with cognitive empathy, statistically significant associations at the level of *p* < 0.001 are observed, showing a positive and direct association (r = 0.403). When the association between preference for Physical Education and sport and empathy is analyzed, statistically significant associations at the level of *p* < 0.05 are observed, showing a positive and direct association in the case of cognitive empathy (r = 0.292), whilst a negative and indirect association is present in the case of affective empathy (r = −287).

When analyzing the association between preference for Physical Education and sport, and evaluation of the subject and of the Physical Education teacher, a statistically significant association at the level of *p* < 0.001 is found, showing a positive and direct relationship (r = 0.247).

In the case of the association between preference for Physical Education and sport, and victimization, the existence of a statistically significant association at the level of *p* < 0.001 is proven, with this being observed as both positive and direct (r = 0.113).

Finally, an association at the level of *p* < 0.05 is found between evaluation of the subject and of the Physical Education teacher, and victimization, with this being negative and indirect (r = −0.065).

## 4. Discussion

The explanatory model on empathy, attitude towards physical education and victimization of adolescents presents a good fit and can serve as a foundation for the detection, prevention and eradication of school violence. Studies along the same lines have been conducted in other parts of the world, with similar results being obtained in Holland [32] and in Spain [33].

The present work and the promotion of empathy in adolescents is one of the current lines of action that should be followed by socio-educational policies in order to encourage healthy lifestyles. In recognition of this, a systematic review of the traits and roles of intimidation was conducted [33]. Higher levels of cognitive empathy can moderate the effect of self-efficacy in adolescents who resist peer pressure when faced with dangerous and deleterious situations [34]. 

High levels of empathy and emotional competence were also associated with a lower risk of belonging to gangs or groups during adolescence. The authors of this study highlight the role of schools in the promotion of emotional regulation, empathy and behavioral regulation within the entire student body, as part of a general strategy to reduce the attraction of students towards joining gangs [35]. Cyberbullying has also been studied and associated with levels of affective and cognitive empathy, showing the sub-components of empathy to be positively and negatively related with cyberbullying [36].

Other recent studies confirm the importance to adolescents of engaging in sporting activities, given the improvements seen in levels of empathy and in the development of personality [37]; even the importance of carrying out mindfulness techniques has been shown [38]. Delinquency has been another topic of research, showing a possible relationship with empathy, although, clearly distinguishing between its cognitive and affective dimensions. More specifically, cognitive empathy is found to correlate with delinquency to a greater extent than with affective empathy [39].

Student’s age also seems to impact the empathy they feel towards the teacher and towards the physical education class. In a study conducted with primary education students (9–12 years old), statistically significant differences were observed, with lower levels of empathy seen amongst the group of students of a more advanced age [40]. 

The study conducted by Di Bartolomeo and Papa [41], also found that individuals exposed to physical activity exhibit greater confidence and pro-social behavior than those who are not exposed to physical activity. These effects are not temporary. 

The data obtained reflects that the indicator exerting the greatest influence amongst adolescents is empathetic joy, whilst empathetic stress exercises a lesser influence. This was a similar occurrence for both sedentary adolescents and those who engaged regularly in physical activity, and can be explained by the age of students who still have low levels of responsibility and for this reason are more predisposed towards happiness in themselves. This has also been proposed by Kwon [37] who adds that club activities contribute to the promotion of a positive personality and help establish emotional empathy. 

In the same way, victimization in both a physical and a verbal sense is more prevalent in students who participate in sport, with this being generated from the very nature of sport itself [18]. On the other hand, the relational dimension is more prevalent in the case of sedentary adolescents. This is characterized by social exclusion of the individual, with the simple fact that not regularly participating in sport produces greater social and relational exclusion. In a cross-sectional cohort study with 991 children and adolescents aged between 7 and 17 years old from a sample of 16 public schools in Barranquilla, Colombia, students who did not regularly engage in physical activity had a greater probability of being victims of violence and demonstrated higher levels of aggression in general. Further, females who did not participate regularly in physical activity reported having less control over their feelings. The results indicate that physical activity must be promoted in schools in order to prevent intimidation and violence [41].

In the present study, no relationship was found between empathy and victimization, these data are in line with those found by other authors who have indicated only a very weak relationship between both concepts [19,42]. In work carried out with 1081 students exposed to the Bystander-High School Curriculum (BITB-HSC) educational program, significant short-term changes were demonstrated in victims [43].

Another study conducted in Korea shows that the results of a teaching method based on empathy had stronger positive effects on the empathy and academic commitment of students than did traditional instruction oriented towards reading. Interviews with students in experimental groups and with the teacher indicated that they were satisfied with the class based on empathy and that they recognized the importance of empathy. This suggests a positive role for a learning model based on empathy within social science students in order to eradicate violence at school [44].

The study establishes a clear relationship between the sedentary behavior of students and cases of victimization. The results indicate that a lack of engagement in physical activity can lead to bullying. These aspects have also been indicated in work developed by Amado-Alonso et al. [45], who signaled that those young people who participate in organized sport have better indices of adaptability and mood state, which in turn reduce stress and disruptive behaviors.

When analyzing the association between the preference for physical education and sports and the assessment of the subject and the Physical Education teacher there is a statistical association, showing a positive and direct relationship that occurs slightly more strongly in the case of adolescents who do not practice physical activity on a regular basis. This can be explained by the fact that adolescents who are physically active and practice physical activity on a regular basis have a lesser influence on the subject of Physical Education, while in sedentary subjects, the expectations and the fun they obtain in this subject will condition them for their preference for Physical Education.

In the case of the association between the preference for Physical Education and sports and victimization, the existence of a statistical association is verified, observing a positive and direct relationship that occurs more strongly in the case of sedentary adolescents. There is also a negative and indirect relationship between the assessment of the subject and the teacher of Physical Education and victimization, which occurs more strongly in the case of adolescents who practice physical activity on a regular basis. No studies have been found that analyze this association, but it is intuited that the practice of physical activity can act as a protective factor against certain forms of violence, although there is no data about the association between attitudes towards physical education and victimization.

The study presents some limitations, the first of these being related with the cross-sectional nature of the study which prevents causal associations from being established. A second limitation is that the data cannot be generalized to the adult population. Another limitation of this study was the selection of the sample, carried out through a convenience sampling process, which may include possible bias.

## 5. Conclusions

The main conclusions of this study dictate that the model based on empathy, attitude towards Physical Education and victimization within adolescents presents a good fit. The relationship between affective empathy and cognitive empathy is positive and direct, however, this relationship is slightly stronger and differentiated in sedentary adolescents than in those who practice physical activity on a regular basis.

Regarding the influence of the indicators of empathy, in the case of cognitive empathy, the indicators influence in a similar way, showing the adoption of perspective of a greater influence than emotional understanding, both in the case of adolescents who practice physical activity on a regular basis and in sedentary adolescents. In the case of affective empathy, the indicator that exerts greater influence is empathic joy, while empathic stress exerts less influence, occurring similarly in both sedentary adolescents and in physical activity practitioners.

Regarding victimization, the verbal dimension is the one that obtained the highest correlation scores, followed by the relational and the physical dimensions. The verbal and physical dimensions have less correlation strength in the case of adolescents who practice physical activity in a habitual way, while the relational dimension is higher in the case of sedentary adolescents.

When analyzing the relationship between empathy and victimization, no association was found with cognitive empathy or affective empathy in any of the groups analyzed. No association has been found to relate the assessment of the subject and the teacher of Physical Education with affective empathy. In the case of the association between the assessment of the subject and the Physical Education teacher with cognitive empathy, in the case of adolescents who practice physical activity on a regular basis there is no association, while in sedentary subjects, there is a positive and direct relationship.

When analyzing the relationship between the preference for physical education and sports and empathy, no statistical association is found in the case of physically active adolescents, while in the case of sedentary adolescents, this association is positive in the case of adolescents who are physically active. This association is positive in the case of cognitive empathy, and negative in the case of affective empathy.

When analyzing the association between the preference for Physical Education and sports and the assessment of the subject and the Physical Education teacher there is a statistical association showing a positive and direct relationship that occurs slightly more strongly in the case of adolescents who do not practice physical activity on a regular basis.

In the case of the association between the preference for Physical Education and sports and victimization, the existence of a statistical association is verified, observing a positive and direct relationship that occurs more strongly in the case of sedentary adolescents.

Finally, there is a negative and indirect relationship between the assessment of the subject and the teacher of Physical Education and victimization, which occurs more strongly in the case of adolescents who practice physical activity on a regular basis.

## Figures and Tables

**Figure 1 ijerph-16-02360-f001:**
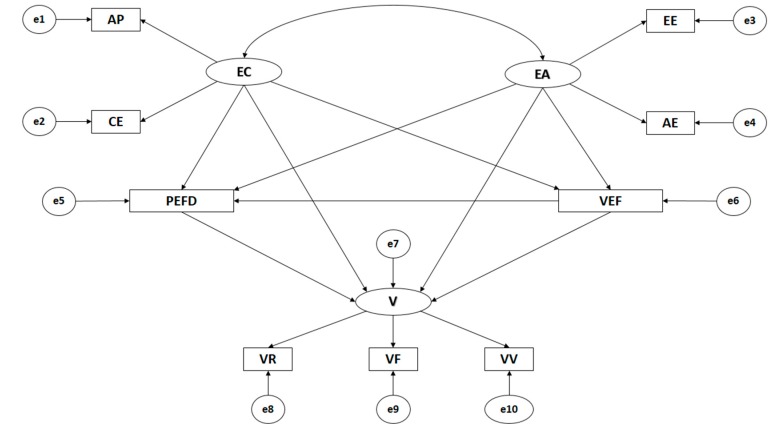
Theoretical model.

**Figure 2 ijerph-16-02360-f002:**
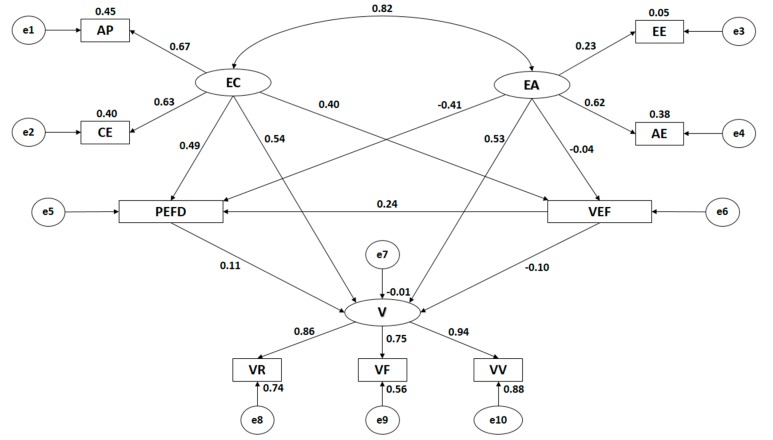
Structural equation model for individuals who regularly engaged in physical activity.

**Figure 3 ijerph-16-02360-f003:**
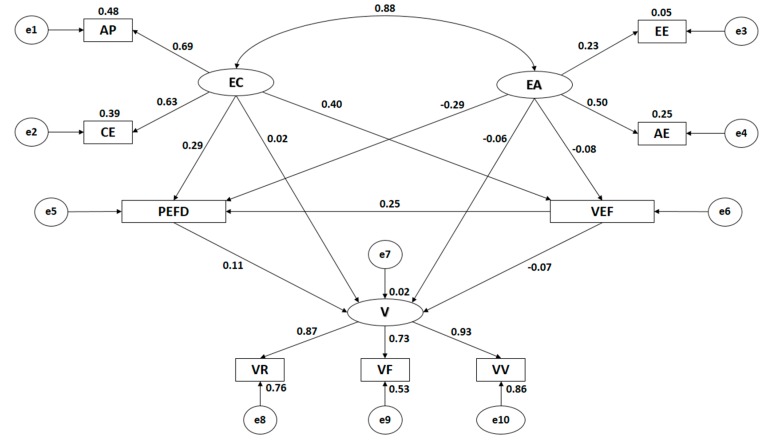
Structural equation model for sedentary participants.

**Table 1 ijerph-16-02360-t001:** Structural model for individuals who regularly engaged in physical activity.

Association between Variables			R.W.				S.W.R.
			Estimates	S.E.	C.R.	P	Estimates
VEF	←	EC	0.649	0.474	1.371	0.170	0.404
VEF	←	EA	−0.184	1.194	−0.154	0.877	−0.045
PEFD	←	EC	0.902	1.083	0.833	0.405	0.489
PEFD	←	EA	−1.932	2.723	−0.709	0.478	−0.410
PEFD	←	VEF	0.271	0.026	10.318	***	0.236
V	←	PEFD	0.105	0.036	2.912	**	0.106
V	←	VEF	−0.111	0.029	−3.847	***	−0.097
V	←	EC	−0.982	1.308	−0.751	0.453	−0.535
V	←	EA	2.498	3.297	0.758	0.449	0.533
CE	←	EC	1.000	-	-	***	0.634
AP	←	EC	1.043	0.042	25.102	***	0.673
EE	←	EA	1.000	-	-	***	0.227
AE	←	EA	2.642	0.241	10.984	***	0.617
VV	←	V	1.000	-	-	***	0.939
VF	←	V	0.697	0.013	53.353	***	0.747
VR	←	V	0.888	0.014	64.176	***	0.861
EC	↔	EA	0.043	0.004	10.633	***	0.821

Note 1: EC, cognitive empathy; EA, affective empathy; AP, perspective adoption; CE, emotional understanding; EE, empathetic stress; AE, empathetic joy; VEF, evaluation of the subject and of the Physical Education teacher; PEFD, preference for Physical Education and sport; V, victimization; VR, relational victimization; VF, physical victimization; VV, verbal victimization. Note 2: R.W., regression weight; S.R.W., standardized regression weight; S.E., standard error; C.R., critical ratio. Note 3: * *p* < 0.05; ** *p* < 0.01; *** *p* < 0.001.

**Table 2 ijerph-16-02360-t002:** Structural model for sedentary participants.

Associations between Variables			R.W.				S.R.W.
			Estimates	S.E.	C.R.	P	Estimates
VEF	←	EC	0.667	0.196	3.400	***	0.403
VEF	←	EA	−0.321	0.436	−0.737	0.461	−0.076
PEFD	←	EC	0.613	0.369	1.662	*	0.292
PEFD	←	EA	−1.528	0.818	−1.867	*	−0.287
PEFD	←	VEF	0.313	0.046	6.823	***	0.247
V	←	PEFD	0.099	0.030	3.342	***	0.113
V	←	VEF	−0.073	0.037	−1.964	*	−0.065
V	←	EC	0.032	0.198	0.164	0.870	0.018
V	←	EA	−0.303	0.437	−0.694	0.488	−0.065
CE	←	EC	1.000	-	-	***	0.626
AP	←	EC	1.137	0.076	14.862	***	0.690
EE	←	EA	1.000	-	-	***	0.229
AE	←	EA	2.272	0.297	7.638	***	0.505
VV	←	V	1.000	-	-	***	0.928
VF	←	V	0.761	0.025	29.945	***	0.728
VR	←	V	0.910	0.025	37.035	***	0.870
EC	↔	EA	0.048	0.006	7.434	***	0.722

Note 1: EC, cognitive empathy; EA, affective empathy; AP, perspective adoption; CE, emotional understanding; EE, empathetic stress; AE, empathetic joy; VEF, evaluation of the subject and of the Physical Education teacher; PEFD, preference for Physical Education and sport; V, Victimization; VR, relational victimization; VF, physical victimization; VV, verbal victimization. Note 2: R.W., regression weight; S.R.W., standardized regression weight; S.E., standard error; C.R., critical ratio. Note 3: * *p* < 0.05; *** *p* < 0.001.

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
