# Peer review of "Association between Level of Empathy, Attitude towards Physical Education and Victimization in Adolescents: A Multi-Group Structural Equation Analysis"

_ijerph, 2019, doi:10.3390/ijerph16132360_

Round 1

Reviewer 1 Report

Dear Authors,

Please find the review in the additional file.

Kind regards

Author Response

Dear the editor and reviewers,

We would like to express our gratitude for the time taken to review this manuscript and for the comments made, which we believe to be critical for producing rigorous and quality research. We have detailed below the changes made in the original article: “Association between level of empathy, attitude towards physical education and victimisation in adolescents: A multi-group structural equation analysis” (ijerph-513075).

Modifications have been made in the original manuscript following the reviewers’ comments. For each modification we have written: the original comment as written by the reviewer in addition to the page and line number; and the change made in response to that comment. Changes have been made using the tool “Track changes” enabling editor and reviewers to identify modifications easily.

Reviewer 1

Comment 1:

In Abstract is given only the aim of the study, short background is needed.

Response 1:

We appreciate the suggestions made since they will improve the quality of the manuscript. A brief introduction on the problems studied has been included in the summary.

Comment 2:

Provided aim of the study is not clear, and do not explain why and what the Authors research for.

Response 2:

Thanks for this indication. In lines 71-75: the following study was proposed with the objective of defining and contrasting an explanatory model of empathy, attitude towards physical education and victimisation in adolescents. A further objective was to analyse through multi-group structural equation analysis, the existing associations between empathy, attitude towards physical education and victimisation as a function of whether or not physical activity was regularly engaged in.

Comment 3:

The lack of clear theoretical background is the fundamental deficiency of this work. Both in introduction and in data analysis in theoretical model design (details below).

Response 3:

Thanks for this indication. This information has been included. Introduction has been restructured.

Comment 4:

The Authors have been using multiple terms:

 victimization, violence, bullying as synonymous concepts – it raises many doubts, especialy in a school context, despite two references cited;

 empathy;

 physical activity, physical education, regularly and not regularly active, regular engagement in physical activity – the terms are mixed, Authors do not provide any explanation why these specific variables were taken into consideration, and also why physical activity engagement is combined/mixed with physical education – totally different concepts and variables?

As well as - how the main variable in MGA – regular engagement in physical activity, was measured and scored?

Response 4:

Thank you very much for this suggestion of improvement. It has proceeded to include these information and correct the errors.

Comment 5:

Under the theoretical models usually lies some strong theory/theories or beliefs come from extensive researcher’s experience in the studied field. This work main weakness is lack of strong theory under the theoretical model design.

Response 5:

Thanks for this indication. Dear reviewer, we have tried to provide a theoretical basis for the development of the model in the introduction.

Comment 6:

The description of results, especially in imputing variables as exo- and endogenous needs reviewing. It’s the consequence of the lack of the theoretical assumptions made on the strong basis.

Response 6:

Thanks for this suggestion of improvement. The introduction has been restructured to give consistency to the document.

Comment 7:

Adopting a unified strategy while building a theoretical model, Authors should follow the bacground methodology and theoty under the used tools/scales. The two latent variables: affective and cognitive empathy – as a result from the two-factor structure of the scale of empathy, should be included into the model indirectly, through another latent variable Empatia.

Response 7:

Thanks for this recommendation. Dear reviewer, each of the two dimensions of empathy is created by the factors that compose it.

Comment 8:

In Results section the final sample size for models is not specified, and is later pointed out as a limitation, therefore it should be known.

Response 8:

The final size of the sample is specified in the sample section, within the methodology. A total of 2,388 adolescents enrolled in compulsory secondary education participated in the present descriptive and cross-sectional study. Ages were reported between 11 and 17 years.

Comment 9:

Nothing is said in the text about missing data and managing missing.

Response 9:

Thanks for this suggestion of improvement. Dear reviewer, in the sample section all the data referring to it, the type of population analyzed, discarded questionnaires, etc. are explained.

Comment 10:

Figures 1. and 2. might be combined to better perception of differences between groups. Data is provided after slashback (ie.0.82/0.79).

Response 10:

Thanks for this indication. The order of the figures is done in this way to follow the storyline, exposing the results of each model, to relate them in the discussion.

Comment 11:

Because there is no theoretical flow through the text, the results description is weak, and gives limited information in the not intelligible form.

Response 11:

Thanks for this indication. The introduction has been restructured to give consistency to the document.

Comment 12:

The same applies to discussions and conclusions, and also the inconsistency in using terms is visible in these two parts of the text.

Response 12:

Thank you very much for this suggestion of improvement. It has proceeded to include these information and correct the errors.

Comment 13:

Conclusions are mostly the repeated results with no clear (with a scientific bacground) reasoning and interpretation of results.

Response 13:

Thanks for this suggestion of improvement. The conclusions of the study have been changed and restructured.

Reviewer 2 Report

The theme of the study is relevant. There is a rationale for the study. The theoretical basis is satisfactory and justifies the study. The objectives are clearly defined and the methodology used is appropriate and consistent with the study proposal. The results and conclusions have good prospects to offer important contributions to the study area. The bibliographic references are current and pertinent to the theme of study.

Author Response

Thank you.

Reviewer 3 Report

This cross-sectional study examined associations between empathy, attitude towards physical education and victimization in Spanish adolescents by using the Structural Equation Modeling. The results showed that several associations between variables of interest were found but there was no relationship between victimization and empathy. The paper has critical concerns.

1.       My major concern is that the rationale of a theoretical model on the relationships between variables of interest is not clear. Although the authors explain that empathy is associated with violence and physical activity contributes to the development of psychosocial factors, the rationale of why empathy may influence attitude towards physical education and in turn victimization is uncertain.

2.       In the same way, why did the authors analyze the relationships between variables of interest as a function of engagement with physical activity? The authors should elaborate on these points.

3.       Relative to the above comment, the authors mentioned that participants were grouped according to those who engage in physical activity or not. But how the authors asked and grouped engagement with physical activity is unexplained.

4.       Some conclusions in the Abstract and the Conclusions section did not support the results of the study. That is, the findings in line 27-29, line 30-31, and line 345-352 are not shown in the Results section.

5.       The authors need to elaborate on the data collection method. What was the study population? How many adolescents responded to the questionnaire? What was the response rate?

6.       This study did not consider any potential confounding factors. The authors should address this point.

7.       What are “P.R.” and “P.E.R.” in Table 1 and Table 2?

Author Response

Dear the editor and reviewers,

We would like to express our gratitude for the time taken to review this manuscript and for the comments made, which we believe to be critical for producing rigorous and quality research. We have detailed below the changes made in the original article: “Association between level of empathy, attitude towards physical education and victimisation in adolescents: A multi-group structural equation analysis” (ijerph-513075).

Modifications have been made in the original manuscript following the reviewers’ comments. For each modification we have written: the original comment as written by the reviewer in addition to the page and line number; and the change made in response to that comment. Changes have been made using the tool “Track changes” enabling editor and reviewers to identify modifications easily.

Reviewer 3

Comment 1:

My major concern is that the rationale of a theoretical model on the relationships between variables of interest is not clear. Although the authors explain that empathy is associated with violence and physical activity contributes to the development of psychosocial factors, the rationale of why empathy may influence attitude towards physical education and in turn victimization is uncertain.

Response 1:

Thanks for this indication. In the introduction section this has been clarified, indicating the possible ralations among the variables, adding victimization in the theoretical model as a part of the premise of empathy as a protective factor in situations of victimization.

Comment 2:

In the same way, why did the authors analyze the relationships between variables of interest as a function of engagement with physical activity? The authors should elaborate on these points.

Response 2:

This topic has been addressed in the introduction. The reason for making a multigroup model in the function of physical activity practice is this relationship with the empathy and victimization, which may influence these variables.

Comment 3:

Relative to the above comment, the authors mentioned that participants were grouped according to those who engage in physical activity or not. But how the authors asked and grouped engagement with physical activity is unexplained.

Response 3:

Thanks for this indication. In the instruments section, the practice of physical activity is explicitly explained.

Comment 4:

Some conclusions in the Abstract and the Conclusions section did not support the results of the study. That is, the findings in line 27-29, line 30-31, and line 345-352 are not shown in the Results section.

Response 4:

Thanks for this suggestion of improvement. The conclusions of the study have been changed and restructured.

Comment 5:

The authors need to elaborate on the data collection method. What was the study population? How many adolescents responded to the questionnaire? What was the response rate?

Response 5:

Dear reviewer, in the sample section all the data referring to it, the type of population analyzed, discarded questionnaires, etc. are explained.

Comment 6:

This study did not consider any potential confounding factors. The authors should address this point.

Response 6:

Dear editor, since validated questionnaires were used for this type of population, no confounding factors have been included. Thanks for your input.

Comment 7:

What are “P.R.” and “P.E.R.” in Table 1 and Table 2?

Response 7:

Thanks for notifying this error. Dear editor, thank you for notifying. It was an error, it had the initials in Spanish and it has been corrected.

Round 2

Reviewer 1 Report

Dear Authors,

My intention when preparing the rewiev of your text was to provide comments to help you to IMPROVE the text, as well as to ensure the proper level of papers accepted to publication in IJERPH.

I'm very sorry that you ignore most of the comments, and do not explain/apply/disscuss them or - i e. see comment 10, your answer and the corrected figures…

Please, see also your corrections in line 55/56 - this is the end or the continuation of the sentense? Line 56-60 this make no sense - first part of the acapite denays the further informations. Line 363 - is this the end of the sentence? Line 386 - small typo - additional dot.

These are just technical glitches, but they are evidence of diligence and rush in working on the text. That's why my final comment is - please go back to the rewiev and rethink your answers to really IMPROVE your work.

Kind regards,

Hanna Nalecz

Author Response

Reviewer 1

Comment 0:

Dear Authors,

My intention when preparing the rewiev of your text was to provide comments to help you to IMPROVE the text, as well as to ensure the proper level of papers accepted to publication in IJERPH.

I'm very sorry that you ignore most of the comments, and do not explain/apply/disscuss them or - i e. see comment 10, your answer and the corrected figures…

Please, see also your corrections in line 55/56 - this is the end or the continuation of the sentense? Line 56-60 this make no sense - first part of the acapite denays the further informations. Line 363 - is this the end of the sentence? Line 386 - small typo - additional dot.

These are just technical glitches, but they are evidence of diligence and rush in working on the text. That's why my final comment is - please go back to the rewiev and rethink your answers to really IMPROVE your work.

Response 0:

First of all we apologize for not responding to all your comments as expected. We respond to the comments again trying to improve the quality of the manuscript.

We apologize for the mistakes made. And if there is still a comment without answering, we ask you to indicate it. Thank you.

We appreciate the suggestions made since they will improve the quality of the manuscript.

Comment 10: Sorry for not correcting it in the previous revision. It was a mistake, thank you very much for letting us know. We have already corrected it. Thank you very much and excuse me again.

The indicated errors have also been corrected. Thank you.

Comment 1:

In Abstract is given only the aim of the study, short background is needed.

Response 1:

We appreciate the suggestions made since they will improve the quality of the manuscript. A brief introduction on the problems studied has been included in the summary.

Comment 2:

Provided aim of the study is not clear, and do not explain why and what the Authors research for.

Response 2:

Thanks for this indication. In lines 71-75: the following study was proposed with the objective of defining and contrasting an explanatory model of empathy, attitude towards physical education and victimisation in adolescents. A further objective was to analyse through multi-group structural equation analysis, the existing associations between empathy, attitude towards physical education and victimisation as a function of whether or not physical activity was regularly engaged in.

Comment 3:

The lack of clear theoretical background is the fundamental deficiency of this work. Both in introduction and in data analysis in theoretical model design (details below).

Response 3:

Thanks for this indication. This information has been included. Introduction has been restructured.

Comment 4:

The Authors have been using multiple terms:

 victimization, violence, bullying as synonymous concepts – it raises many doubts, especialy in a school context, despite two references cited;

 empathy;

 physical activity, physical education, regularly and not regularly active, regular engagement in physical activity – the terms are mixed, Authors do not provide any explanation why these specific variables were taken into consideration, and also why physical activity engagement is combined/mixed with physical education – totally different concepts and variables?

As well as - how the main variable in MGA – regular engagement in physical activity, was measured and scored?

Response 4:

Thank you very much for this suggestion of improvement. It has proceeded to include these information and correct the errors.

Comment 5:

Under the theoretical models usually lies some strong theory/theories or beliefs come from extensive researcher’s experience in the studied field. This work main weakness is lack of strong theory under the theoretical model design.

Response 5:

Thanks for this indication. Dear reviewer, we have tried to provide a theoretical basis for the development of the model in the introduction.

Comment 6:

The description of results, especially in imputing variables as exo- and endogenous needs reviewing. It’s the consequence of the lack of the theoretical assumptions made on the strong basis.

Response 6:

Thanks for this suggestion of improvement. The introduction has been restructured to give consistency to the document.

Comment 7:

Adopting a unified strategy while building a theoretical model, Authors should follow the bacground methodology and theoty under the used tools/scales. The two latent variables: affective and cognitive empathy – as a result from the two-factor structure of the scale of empathy, should be included into the model indirectly, through another latent variable Empatia.

Response 7:

Thanks for this recommendation. Dear reviewer, each of the two dimensions of empathy is created by the factors that compose it.

Comment 8:

In Results section the final sample size for models is not specified, and is later pointed out as a limitation, therefore it should be known.

Response 8:

Dear editor, the requested information has been included at the beginning of the results.

Comment 9:

Nothing is said in the text about missing data and managing missing.

Response 9:

Thanks for your contributions, which undoubtedly improve the quality of the manuscript. It has been included in the description section of the sample, the figures of the study participants, the questionnaires answered correctly, the ones eliminated, the response rate and the selection criteria of the sample.

Comment 10:

Figures 1. and 2. might be combined to better perception of differences between groups. Data is provided after slashback (ie.0.82/0.79).

Response 10:

Thanks for this indication. Sorry for not correcting it in the previous revision. It was a mistake, thank you very much for letting us know. We have already corrected it. Thank you very much and excuse me again.

Comment 11:

Because there is no theoretical flow through the text, the results description is weak, and gives limited information in the not intelligible form.

Response 11:

Thanks for this indication. The introduction has been restructured to give consistency to the document.

Comment 12:

The same applies to discussions and conclusions, and also the inconsistency in using terms is visible in these two parts of the text.

Response 12:

Thank you very much for this suggestion of improvement. It has proceeded to include these information and correct the errors.

Comment 13:

Conclusions are mostly the repeated results with no clear (with a scientific bacground) reasoning and interpretation of results.

Response 13:

Thanks for this suggestion of improvement. The conclusions of the study have been changed and restructured.

Reviewer 3 Report

1, 2.         The authors’ explanation about the rationale of the relationships between variables of interest is insufficient. The authors mentioned that empathy is associated with violence and physical activity contributes to the development of social factors. This implication may be that physical activity produces empathy and in turn, reduces violence. However, the theoretical model showed paths such as empathy --> attitudes toward physical education --> violence. There are inconsistent directions. In addition, I cannot understand why the authors used attitudes towards physical education instead of physical activity, although the authors mentioned the role of physical activity in the development of social factors and violence prevention.

3.             The rationale of the physical activity category is unclear. Why is 3 or more hours of physical activity a week?

4.             It is still remained descriptions not supported by the results. That is, the findings in lines 32-34, lines 338-339, and lines 343-344 are not shown in the Results section. The authors did not compare the levels of victimization and attitudes toward physical education between physically active students and sedentary students.

5.             The authors don’t respond to this point sufficiently. What was the study population? How many adolescents responded to the questionnaire? What was the response rate? Is there any selection bias?

6.             The authors don’t address this point in the text. At least, the authors should mention as a limitation.

Author Response

Reviewer 3

Comment 1:

1, 2. The authors’ explanation about the rationale of the relationships between variables of interest is insufficient. The authors mentioned that empathy is associated with violence and physical activity contributes to the development of social factors. This implication may be that physical activity produces empathy and in turn, reduces violence. However, the theoretical model showed paths such as empathy --> attitudes toward physical education --> violence. There are inconsistent directions. In addition, I cannot understand why the authors used attitudes towards physical education instead of physical activity, although the authors mentioned the role of physical activity in the development of social factors and violence prevention.

Response 1:

Dear reviewer, thank you very much for your comment. The relationships of the proposed model are correlations, so they occur in both directions although the arrows are unidirectional, if one variable increases the other also does, and if one decreases, the other decreases. Regarding the inclusion of the practice of physical activity, it was considered appropriate to observe how the aptitudes that the students have towards the subject of Physical Education work. In future work we will include the practice of physical activity as you recommend. We appreciate your comment.

Comment 2:

3. The rationale of the physical activity category is unclear. Why is 3 or more hours of physical activity a week?

Response 2:

Thanks for your improvement comment. The indications provided by global organizations such as WHO have been used, indicating that the minimum recommended physical activity is three hours a week.

Comment 3:

4. It is still remained descriptions not supported by the results. That is, the findings in lines 32-34, lines 338-339, and lines 343-344 are not shown in the Results section. The authors did not compare the levels of victimization and attitudes toward physical education between physically active students and sedentary students.

Response 3:

Thanks for this indication. We have proceeded to eliminate the comments that were not based on the results and comparisons of the data indicated in the discussion have been made, although no research has been found with which to compare the data obtained.

Comment 4:

5. The authors don’t respond to this point sufficiently. What was the study population? How many adolescents responded to the questionnaire? What was the response rate? Is there any selection bias?

Response 4:

Thanks for your contributions, which undoubtedly improve the quality of the manuscript. It has been included in the description section of the sample, the figures of the study participants, the questionnaires answered correctly, the ones eliminated, the response rate and the selection criteria of the sample.

Comment 5:

6. The authors don’t address this point in the text. At least, the authors should mention as a limitation.

Response 5:

Thank you very much for your comment. It has been included in the limitations of the investigation explaining that the sampling done is for convenience and a population bias can be found.